# Synthesis of Zn-Saponite Using a Microwave Circulating Reflux Method under Atmospheric Pressure

**Bing-Sheng Yu [1],*, Wei-Hsiang Hung [1], Jiann-Neng Fang [2] and Yu-Ting Yu [3]**

[1] Institute of Mineral Resources Engineering, National Taipei University of Technology, Taipei 10608, Taiwan; 410154009@ems.ndhu.edu.tw

[2] Department of Research, National Taiwan Museum, Taipei 10046, Taiwan; jnfang@ntm.gov.tw

[3] Geotechnical Engineering Department, Sinotech Engineering Consultants, Ltd., Taipei 10570, Taiwan; yuting.geo@gmail.com

* Correspondence: bing@ntut.edu.tw or bing3333@gmail.com; Tel.: +886-2-2771-2171

**Abstract:** Smectite is a common clay mineral in nature. Due to its tendency to swell and its strong cation exchange capacity (CEC), smectite is prevalently used in industrial and technological applications. Numerous scholars have explored smectite synthesis, which usually involves autoclaving under high pressure. However, this approach requires an array of expensive equipment, and the process consumes time and energy. This study adopted self-developed equipment to synthesize zinc saponite (Zn-saponite), a type of trioctahedral smectite, using a microwave circulating reflux method under atmospheric pressure. Compared with the conventional hydrothermal methods, the proposed method entails fewer constraints regarding the synthesis environment and can be more easily applied to large-scale synthesis. The phase purity of the synthetic product was examined using X-ray diffraction and the CEC of the product was tested. The results revealed that the microwave circulating reflux method could synthesize Zn-saponite in 16 h under atmospheric pressure, and the CEC of the product reached 120 cmol(+)/kg. In addition, the product exhibited larger basal spacing and a 32% increase in CEC compared with Zn-saponite synthesized using a hot-plate under atmospheric pressure.

**Keywords:** Zn-saponite; microwave circulating reflux; synthesis

## 1. Introduction

Smectite is a clay mineral commonly found in nature and an essential ceramic material [1]. Due to its unique properties, smectite is prevalently used in various industries, including oil drilling, metallurgy, construction, chemical engineering, food manufacturing, aquaculture, and environmental engineering. However, its chemical composition, uneven microporous structure, and high concentration of impurities impede its use in high-level applications such as organo-modifications and catalytic reactions [2]. Although naturally formed smectite can undergo purification pretreatment, such processes are complex, expensive, and time-consuming, and result in environmental pollution. In addition, the resulting products contain impure phases. Synthetic smectite exhibits an even composition and structure, and its physical and chemical properties can be controlled during synthesis. Accordingly, numerous scholars have explored smectite synthesis [2–5].

According to its crystalline structure, smectite is categorized into trioctahedral smectite (including hectorite, saponite, and sauconite) and dioctahedral smectite (including montmorillonite, beidellite, and nontronite). Researchers have attempted to synthesize smectite of all types, but these synthesis processes have mostly been performed using hydrothermal methods under high temperatures. For example, saponite is typically synthesized under a hydrothermal condition of 5–17 MPa and 150–500 °C [6–8] by

using specialized equipment. Conventional hydrothermal methods involve placing a reactor inside an autoclave, where the energy is transferred inward to the reactants. However, this synthesis process consumes considerable amounts of energy.

The development of microwave synthesis enables direct heating inside a reactor, which facilitates completing the reaction under a lower temperature and in a shorter time [9–11]. For example, Trujillano's group successfully synthesized saponite and saponite-like materials by using microwave irradiation at 180 °C [11–14]. Vicente et al. [15] used the microwave hydrothermal method at 120 °C to synthesize hectorite in 8 h. The resulting product exhibited a cation exchange capacity (CEC) of 68 cmol(+)/kg, and the synthesis time was notably shorter than the 192-h reaction time required for conventional hydrothermal methods. Vicente et al. [16] also adopted the microwave hydrothermal method at 180 °C to synthesize saponite in 6 h. The product exhibited a CEC of 65 cmol(+)/kg.

Conventional hydrothermal methods entail using a specialized autoclave. Although microwave hydrothermal methods can reduce the reaction temperature and time, it also has certain drawbacks. Previous studies adopting such methods to synthesize smectite have used temperatures higher than 100 °C, and the reaction must also be conducted inside an autoclave. Such methods involve substantial equipment requirements and are difficult to apply in large-scale production. Therefore, developing alternative synthesis methods to produce smectite in large batches is imperative.

Of various alternative, nonhydrothermal methods (i.e., synthesis under the atmospheric pressure) have seen the most development [17]. Klopproge et al. [18] reported several cases of smectites synthesis below 100 °C. Vogels et al. [17] prepared saponites at 90 °C and 1 atmosphere over several hours. Prikhod'ko et al. [19] used a hot-plate under atmospheric pressure to synthesize saponite-like materials at 90 °C. Other researchers also synthesized Zn-saponite [20,21] and stevensite [22] under atmospheric pressure.

Scholars have used microwave hydrothermal methods to synthesize saponite in autoclaves or autoclave-like containers, but not under atmospheric pressure. In addition, no study has used microwave-assisted methods to synthesize saponite under atmospheric pressure. Therefore, this study developed a microwave circulating reflux method to produce Zn-saponite under atmospheric pressure. The phrase purity and CEC of the product are discussed herein.

## 2. Experimental Methods

For comparison, reagents used for synthesizing Zn-saponite were based on Prikhod'ko et al. [19]. $Na_{1.2}[Zn_6](Si_{6.8}Al_{1.2})\cdot O_{20}(OH)_4\cdot nH_2O$ was used to prepare the precursor. The final precursor was a mixture of precursors A and B, where 1.1 L of final precursor was a mixture of 600 mL of precursor A and 500 mL of precursor B. For precursor A, an $Al(OH)_4$ solution was prepared by dissolving 4.22 g of $Al(NO_3)_3\cdot 9H_2O$ (99%) in 28 mL of 2 M NaOH (99%). This solution was then added to another solution containing 30.0 g of $Na_2SiO_3\cdot 9H_2O$ (99%) in 75 cm$^3$ of deionized water under vigorous stirring. The resulting gel was kept at room temperature for 1 h without stirring. Next, the gel was loaded into deionized water, to a volume of 600 mL, and mixed evenly. For precursor B, 500 mL of a separately prepared aqueous solution containing 32.60 g of $Zn(NO_3)_2\cdot 6H_2O$ (99%) and 0.80 mol of $CO(NH_2)_2$ (98%) in deionized water. According to Prikhod'ko et al. [19], urea can be used as a hydrolyzing agent because of its thermally initiated decomposition, which makes it possible to smoothly vary the pH of the reaction medium, and thus, its hydrolysis rate when forming the target material can be controlled. The two precursors (A and B) were evenly mixed to acquire the final precursor.

This study self-developed an atmospheric pressure microwave circulating reflux device (Figure 1) consisting of three microwave reaction chambers (with a total power of 100 w) and a reaction tube with a diameter of 1 cm. The initial precursor volume was set as 1 L and added in the flask. A peristaltic pump was used to transfer the precursor from the near bottom of the flask into the microwave device and then back to the flask. To prevent precipitation of saponite products, the pump was used to maintain a circulating flow at 1200 mL/min. Although there was a temperature maintenance device under the flask, the temperature in the flask was mainly maintained using the hot liquid refluxed

by the microwave reaction. However, the residual temperature in the flask may have also promoted saponite formation. Therefore, in this study, a control experiment using only a hot plate was performed for comparison.

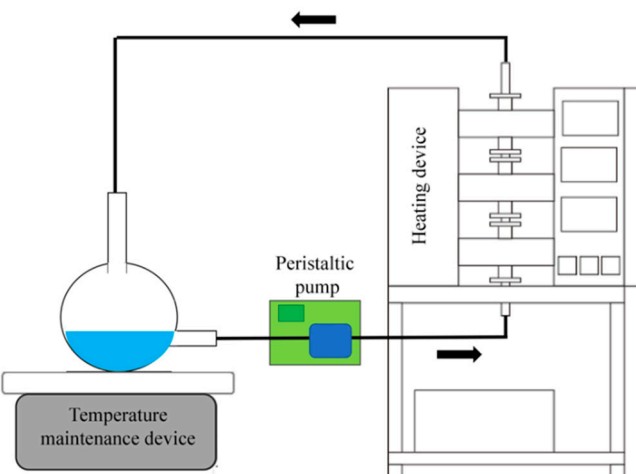

**Figure 1.** Atmospheric pressure microwave reflux device for Zn-saponite synthesis.

The temperature of the microwave device outlet was maintained at about 95 °C. After the reacting solution passed through the outlet, the solution temperature gradually decreased and remained at approximately 90 °C. The reaction times were set as 16 and 24 h. To achieve significantly different results, the experimental conditions were set to change considerably. Other synthesis processes were performed using the same microwave power but with half the precursor volume (0.5 L) and using the same precursor volume but with the microwave power reduced by half (total: 50 w). The resulting products were compared.

To synthesize the control sample, this study adopted the method used in Prikhod'ko et al. [19]; a hot-plate was used to synthesize smectite. In the control experiment, the reaction temperature of the precursor in the flask was set at 95 °C.

After the reaction was complete, the products were washed three times with water, separated centrifugally, and then, placed in a 70 °C oven. The dried product underwent X-ray diffraction (XRD) to verify its crystalline phase. XRD analysis of the synthesized samples was conducted using a D2-Phaser powder diffractometer (Bruker, Germany, Cu-Kα radiation), with 2θ of 5–70°, step size of 0.02°, and scanning rate of 3°/min. For the quantitative analysis of phase purity, part of each product was taken, added to 10% alumina powder as an internal standard, mixed well, and then, measured through XRD.

According to Yu and Liu [23], a software environment that combines a Profex graphical user interface with the software BGMN can handle models for structural disorder of smectites [24]; this software environment was thus selected for the Rietveld refinement quantitative phase analysis of the synthetic saponite samples. In addition, the mineralogical compositions of impurities were determined through the Rietveld refinement of the synthetic smectite products. The software simulated and calculated repeatedly until the content of alumina powder in the composition was close to 10% to obtain an accurate content of each phase. After the alumina content was deducted, the true content of each mineral phase in the original product was converted.

The method discussed by Bergaya and Vayer [25] was used to measure the CEC.

## 3. Results and Discussion

### 3.1. Crystalinity and Phase Purity

Some synthetic products were subjected to saponite confirmation experiments. In the ethylene glycol experiment, d(001) of the products expanded to approximately 1.82 nm, and when heated to

400 °C d(001) shrunk to approximately 0.98 nm. Furthermore, with CTAB (cetyltrimethylammonium bromide or hexadecyl trimethyl ammonium bromide) intercalation, its d(001) could be increased to 1.92–2.03 nm. These experimental results suggested that the synthesized product belongs to the smectite mineral group.

Figure 2 displays the XRD pattern of the products obtained using the two methods. When the hot-plate method [19] was applied for 16 h, the crystalline phase of the product (Hot-plate-16) was slightly noticeable. The 2θ pattern displayed only three faint peaks at approximately 28°, 35°, and 60°. When the synthesis time was extended to 24 h, the XRD pattern of the product (Hot-plate-24) exhibited more noticeable diffraction peaks of Zn-saponite.

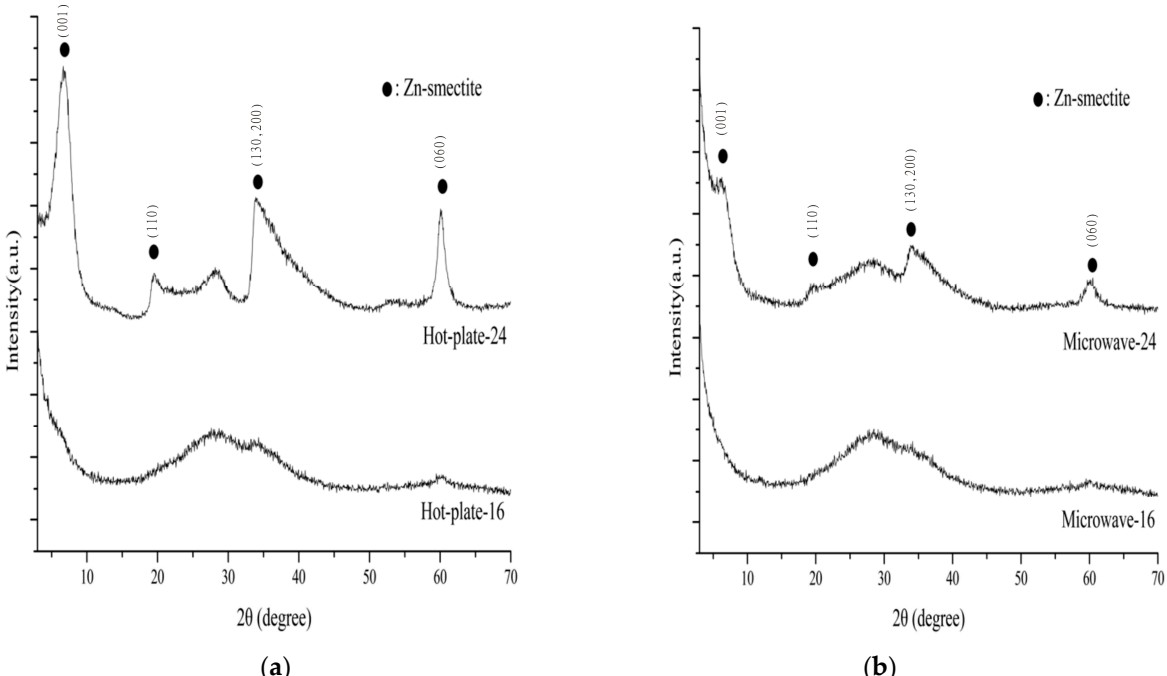

**Figure 2.** XRD pattern of products synthesized using a hot-plate under atmospheric pressure (**a**) and by using the microwave reflex method (**b**).

When the atmospheric pressure microwave reflex method was employed for 16 h (Microwave-16), the XRD pattern did not exhibit noticeable diffraction peaks. When the reaction time was extended to 24 h (Microwave-24), despite the appearance of Zn-saponite diffraction peaks, the peak intensity was lower than that of the product synthesized using the hot-plate (Figure 2).

The basal spacing ($d_{001}$) of Hot-plate-24 was 1.298 nm at ambient humidity, slightly smaller than that of Microwave-24 (1.325 nm). All D-values of the (060) reflections were 0.1539 nm, suggesting that trioctahedral saponite was synthesized successfully [26,27].

After the addition of 10% alumina powder as an internal standard, phase purity analyses of the products were conducted for the two synthesis methods for 24 h on BGMN. The results revealed that the product synthesized using the hot-plate exhibited a phase purity of 97.3%. By contrast, the product synthesized using the microwave circulating reflux method only exhibited a phase purity of 45.0% (Table 1). In addition to saponite, the remaining constituents of the product were only amorphous, without other impurity phases.

**Table 1.** Phase purity and basal spacing of Zn-saponite synthesized using various methods.

| Sample Number | Phase Purity (%) | Basal Spacing (nm) | $d_{060}$ (nm) |
|---|---|---|---|
| Hot-plate-24 | 97.3 | 1.298 ± 0.003 | 0.1539 ± 0.0004 |
| Microwave-24 | 45.0 | 1.325 ± 0.003 | 0.1539 ± 0.0004 |
| Microwave-24-0.5E | 50.2 | 1.303 ± 0.003 | 0.1538 ± 0.0004 |
| Microwave-24-0.5V | 43.9 | 1.383 ± 0.003 | 0.1539 ± 0.0004 |

The amount of the crystallized phase, determined through XRD pattern (Figure 2), and the calculated phase purity (Table 1) indicated that the crystalline quality of the products synthesized using the microwave circulating reflux method were inferior to the products synthesized using a hot-plate.

Although many studies have reported that microwave reactions can substantially reduce the synthesis time of a mineral or material, some studies have demonstrated that microwave radiation occasionally damages the crystalline structure of a synthetic mineral or material during synthesis. For example, Girnus et al. [28] noted that microwave radiation might destroy H-bridges, resulting in rapid gel dissolution and the formation of Al–O–P bonds during $AlPO_4$ synthesis. During the synthesis of zinc oxide nanorods, Zhu et al. [29] discovered that a too high or too low working microwave power will destroy crystal structures. Zhang et al. [30] indicated that the crystalline structures of zeolite were destroyed when microwave power was above 480 W. Fan et al. [31] suggested that microwave heating tends to destroy the lamellar structure within starch granules before the disordering of the double helixes within amylopectin due to its rapid heating rate.

The microwave device used in this study had only three small chambers, each had a reaction space with a height of 10 cm and an inner diameter of 1 cm. The volume of the total reaction space was only 23.6 mL. This means that if the reactant volume is 1000 mL, the reactant only undergoes a microwave reaction for 2.36% of the entire reaction time. Consequently, despite the reaction time being 24 h, the precursor only underwent a microwave reaction for approximately 0.57 h because of the limited space in the reaction chambers. However, in this short period of time, the precursor is exposed to strong microwave radiation. The excessively strong energy emitted by the microwave device resulted in Zn-saponite having unfavorable phase purity and crystallinity.

To further explore the effect of microwave radiation on the crystalline structure of synthetic Zn-saponite, this study reduced either the precursor volume (Microwave-24-0.5V) or microwave power by half (Microwave-24-0.5E) and compared the obtained synthetic products. When the volume of the solution is halved, it circulates at a double rate, and thus its exposure time under microwave irradiation is double. If the supposition regarding the negative effect of overly strong microwave radiation on Zn-saponite crystallinity is valid, reducing the precursor volume by half would result in less crystalized products because the exposure time of precursor to microwave radiation was increased. This would also reduce the XRD peak intensity. By contrast, reducing the microwave power by half would reduce the energy of the microwave reaction, resulting in more crystalized products with stronger XRD peak intensities.

Figure 3 presents the XRD pattern of Microwave-24, Microwave-24-0.5V, and Microwave-24-0.5E. The results revealed that Microwave-24-0.5V, which was exposed to microwave radiation for the longest duration, exhibited the least noticeable peak in the basal plane (001), indicating unfavorable crystallinity. Additionally, this sample exhibited a slightly lower phase purity (43.9%) than did Microwave-24 (Table 1) and had an increased basal spacing of 1.383 nm.

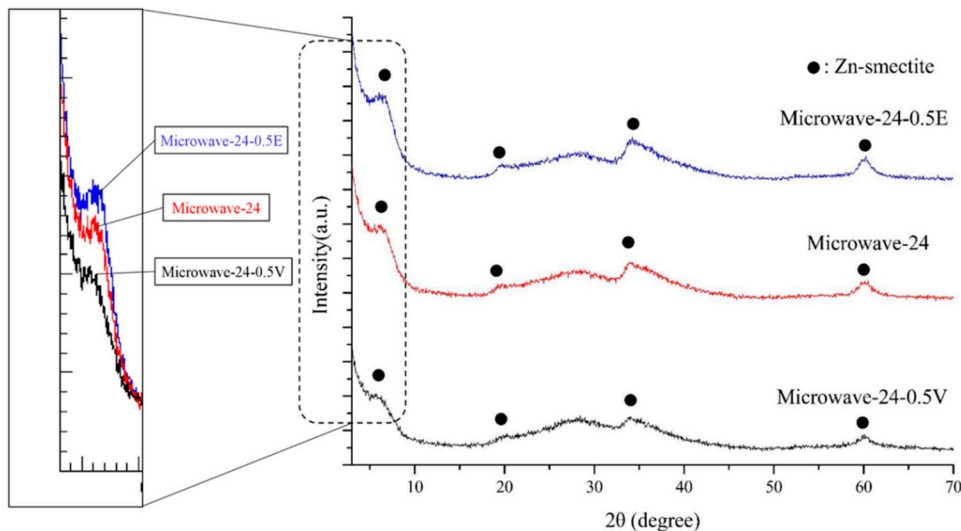

**Figure 3.** XRD pattern of Zn-saponite synthesized using the microwave circulating reflux method with different parameters.

The basal plane (001) peak of Microwave-24-0.5E was stronger than that of Microwave-24, indicating favorable crystallinity. This sample also exhibited slightly higher phase purity (50.2%) than did Microwave-24 and a slightly reduced basal spacing of 1.303 nm.

The aforementioned results verified that although the proposed device can be used to synthesize Zn-saponite in 24 h, during which the solid phase is exposed to microwave for only 0.57 h, the overly strong microwave energy produced by the device can damage the crystalline structure of the synthetic product, resulting in unfavorable phase purity and crystallinity.

## 3.2. Cation Exchange Capacity

The cation exchange capacity (CEC) of smectite is a critical property affecting its applicability. Under atmospheric pressure, this study used a hot-plate to heat the precursor for 16 and 24 h to synthesize smectite, resulting in CEC of 84 and 91 cmol(+)/kg, respectively (Table 2). When the microwave circulating reflux method was adopted to synthesize smectite under atmospheric pressure, the CEC of the product was higher than that synthesized using the hot-plate (Hot-plate-24) regardless of the reaction time, reaching 96 and 105 cmol(+)/kg, respectively. Reducing the microwave power or precursor volume by half also resulted in Zn-saponite (Microwave-24-0.5E and Microwave-24-0.5V) having a higher CEC, reaching 112 and 120 cmol(+)/kg, respectively.

**Table 2.** Cation exchange capacity (CEC) of Zn-saponite synthesized using various methods.

| Sample Number | Reaction Time (h) | CEC (cmol(+)/kg) |
|---|---|---|
| Hot-plate-16 | 16 | 84 |
| Hot-plate-24 | 24 | 91 |
| Microwave-16 | 16 | 96 |
| Microwave-24 | 24 | 105 |
| Microwave-24-0.5E | 24 | 112 |
| Microwave-24-0.5V | 24 | 120 |

The permanent layer charge of synthesized saponites according to their theoretical structural formulas was calculated to be 1.20 $\zeta_{TOT}$ (per $O_{20}(OH)_4$). Therefore, in this study, the highest CEC of pure saponite products could theoretically be 120 cmol(+)/kg. However, the saponite purity of the microwave-24–0.5-V sample was only 43.9%, and the measured CEC was 120 cmol(+)/kg. If the CEC of amorphous material on the product is ignored, the CEC of saponite in the product might be as high as

270 cmol(+)/kg—indicating that the CEC of the saponite product is not entirely due to the permanent layer charge, but that a large part of it might be from the surface charge origin.

Numerous scholars have reported that crystal defects can increase the CEC of a mineral. For example, Murray and Lyons [32] stated that the differences in the CEC of various clay minerals have been attributed to differences in the degree of crystallinity. Ormsby et al. [33] explored the exchange behavior of kaolines with various degrees of crystallinity and mentioned that the CEC was well correlated with the surface areas that were affected by changing from well crystallized to poorly crystallized kaolines. Peigneur et al. [34] asserted that broken bonds on the edges of montmorillonite crystals affect ion exchange behavior. Siantar and Millman [35] demonstrated that CEC could be correlated with the number of defects created in the zeolite structure.

The XRD results indicated that Zn-saponite synthesized using the microwave circulating reflux method exhibited weaker peaks than did that synthesized using the hot-plate method, indicating that it had unfavorable crystallinity. However, the proposed method could create synthetic products with higher CEC levels.

## 4. Conclusions

This study successfully adopted the microwave circulating reflux method at approximately 90 °C and under atmospheric pressure to synthesize Zn-saponite in 24 h. Compared with the product obtained using the hot-plate, Zn-saponite synthesized using the proposed method exhibited larger basal spacing, which can be conducive to subsequent property modification through intercalation. Although Zn-saponite synthesized using the proposed method exhibited unfavorable crystallinity, its CEC was increased by approximately 32% compared with that of the product from the hot-plate method, reaching 120 cmol(+)/kg. Compared with conventional hydrothermal methods, the microwave circulating reflux method has fewer constraints regarding the synthesis environment and can be easily applied in large-scale production. Therefore, it holds considerable potential for synthesizing smectite with high applicability.

**Author Contributions:** Conceptualization and Project administration, B.-S.Y.; Methodology, B.-S.Y. and J.-N.F.; Validation, W.-H.H.; Formal Analysis, W.-H.H.; Writing—Original Draft Preparation, B.-S.Y. and Y.-T.Y.; Writing—Review and Editing, B.-S.Y., J.-N.F. and Y.-T.Y. All authors have read and agreed to the published version of the manuscript.

**Funding:** This research was funded by the Ministry of Science and Technology of the Republic of China, grant number MOST-106-2116-M-027-003.

**Conflicts of Interest:** The authors declare no conflict of interest. The sponsors had no role in the design, execution, interpretation, or writing of the study.

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
