# Peer review of "Synthesis of Zn-Saponite Using a Microwave Circulating Reflux Method under Atmospheric Pressure"

_minerals, doi:10.3390/min10010045_

Round 1

Reviewer 1 Report

This manuscript reports some interesting results, and may become acceptable for publication in Minerals. However, it must be strongly revised before definitive acceptance.

Introduction.

Line 43: “bi-octahedral” must be “dioctahedral”. Please, see the Handbook of Clay Science, Elsevier, 2013.

More credit may be given to the work from Bisio group on synthesis of saponite, citing at least these papers:

Bisio, C., Gatti, G., Boccaleri, E., Marchese, L., Superti, G.B., Pastore, H.O., Thommes, M., 2008. Understanding physico-chemical properties of saponite synthetic clays. Microporous Mesoporous Mater. 107, 90–101.

Costenaro, D., Gatti, G., Carniato, F., Paul, G., Bisio, C., Marchese, L., 2012. The effect of synthesis gel dilution on the physico-chemical properties of acid saponite clays. Microporous Mesoporous Mater. 162, 159–167.

And also, and specially, to the work from Trujillano group, as this group synthesize saponite using microwave irradiation, citing at least these papers:

Trujillano, E. Rico, M.A. Vicente, V. Rives, L. Bergaoui, S. Ben Chaabene, A. Ghorbel. Microwave-Assisted Synthesis of Fe3+ Saponites. Characterization by X-Ray Diffraction and FT-IR Spectroscopy. Macla, 11, 189-190, 2009.

Trujillano, E. Rico, M.A. Vicente, M. Herrero and V. Rives. Microwave radiation and mechanical grinding as new ways for preparation of saponite-like materials. Applied Clay Science, 48, 32-38, 2010.

Trujillano, E. Rico, M.A. Vicente, V. Rives, K.J. Ciuffi, A. Cestari, A. Gil, S.A. Korili. Rapid microwave-assisted synthesis of saponites and their use as oxidation catalysts. Applied Clay Science, 53, 326-330, 2011.

Trujillano, E. Rico, M.A. Vicente, V. Rives, I. Sobrados, J. Sanz. Saponites containing divalent transition metal cations in octahedral positions – Exploration of synthesis possibilities using microwave radiation and NMR characterization. Applied Clay Science, 115, 24–29, 2015.

You may also note than in the last paper from this group, that in fact you cite as reference number 8 in your manuscript, a saponite containing Zn2+ is reported. So, your sentence in lines 71-73 may be modified accordingly. You write “no studies have used microwave methods to synthesize zinc saponite (Zn-saponite) under atmospheric  pressure” The sentence is correct, as previously the synthesis have been carried out in microwave closed reactors, but reading your work one thinks that this is the first synthesis of Zn-saponite, so I suggest to clearly explain the novelty of your work.

Experimental.

Line 78: In the general formula of saponite, it is written that the exchange cations are (Mz+)x/z, but then it is indicated that the composition of the octahedral and tetrahedral sheets is [Zn6](Si6.8Al1.2). Well, with this composition of the sheets, the charge of the layer should be 1.2 units per formula, and if the compensating cation has a charge Mz+, the required amount should be (1.2)/z. Besides, I think that the only exchangeable cation present should be sodium, so the interlayer cations should be simply written as Na1.2.

I do not find evident the composition of the solutions. One solution contains zinc nitrate and urea (line 82). Why this composition? Urea is usually used because its hydrolysis, easily controlled by heating, allows to increase progressively the pH of a solution. But the other solution is strongly alkaline, as it contains 2 M sodium hydroxide (line 79). Both solutions, although prepared separately, are mixed before the beginning of the synthesis. So, the function of urea is difficult to understand. Please, explain carefully.

Page 3. I do not see clearly how the circulation of the solutions is done (in fact, it may be more correct to talk of suspension, or at least solution/suspension, as the solid may precipitate from the beginning of the process). The authors use a peristaltic pump to circulate the solution (about this, is it correct the velocity of 1200 mL/min indicated in lines 87-88?, I think that this velocity is excessive). In the first paragraph in page 5, the authors calculate the parameters when the volume of the solution is 1 L (the time the solution is under irradiation…). In lines 94-96 the authors say that “Other synthesis processes were performed using the same microwave power but with half the precursor volume (0.5 L) and using the same precursor volume but with the microwave power reduced by half (total: 50 w)”. However, I think that such a comparison is very difficult when strongly changing the synthesis conditions. If the volume of the solution is the half, how does this affect the circulation? Does it circulate at the double of velocity, and thus is it expose to microwave irradiation the double of time?

When doing a microwave assisted synthesis, the temperature is usually controlled by controlling the irradiation and by measuring the temperature inside the solution. Here, the authors fix the temperature of the solution by an external bath, and when submerged in this bath the temperature of the mixture is 95ºC. But what happened when the solution is circulating by the microwave oven, during the time that it is inside the furnace? The only parameter illustrating this is the power of the microwave irradiation, but this gives poor information about the temperature.

Results.

The authors write several times “XRD spectra”. Well, X-ray diffraction is, obviously, a diffraction technique, not a spectroscopic technique, so the use of the word spectra is not correct. The authors can say XRD patterns, XR diffractograms, …

The use of alumina as internal standard to calculate the purity should be explained. I mean, how were the reported values obtained after register the diffractogram of the samples containing 10% alumina? Some solids show well-defined peaks, and other solids show very wide and not-resolved peaks. How does this affect the measurement?

English style should be revised. Also, small typographic mistakes are found, as “miceowave” instead of “microwave” (line 137) or “mintmorillonite” instead of “montmorillonite” (line 186).

Reviewer 2 Report

The idea to use a microwave circulating reflux method is good, further inviestigation are needed to prove the formation of Zn-saponite, many experimentals details as well as characterization of the different samples are missing, I therefore recommend major revisions of this manuscript and the use of chemical and thermal analysis as well as of FTIR beside XRD and CEC measurements carried out by the authors.

here are the main points to be adressed :

the introdction should be modified as it is not informative enough for the readers and as there are many mistakes ( line 43 : dioctahedral should be written instead of bi-octahedral, line 47 the authors should chose between Kelvin and celsius degrees, lithim-bentonite is not a correct name. The authors states that hydrothermal methods require expensive specialized equipment and that the synthesis duration are long, this is not true as some autoclaves are low cost, that this method is widely used at industrial scale anas that the crystallization duration can be as short as 4h under autogenous pressure.

line 69 the authors states that neither Vogels et al. neither Prikhod'ko analyzed the phase purity, this has to be checked in the publication.

the experimental part lacks of details, such as the purity of the reactant, the molar composition of the mixture , the temperature at which the reactants are added, the volume of precursord A and B,the voume of water (distilled or not?) used to wash  the obtained solid, the recovering method ( centrifugation or filtration), the duration of the drying step, etc...the description of the microwave device shouls also be more details.

It is also quite difficult for the reader to understand why the reaction times were set at 16 and 24 hours, why the authors used half the precursor volume and also kept the microwave power reduced by half. Nothing is said about the equipment used to collect XRD data, about the sample preparation this is required.

the part dedicate to the results and discussion lacks on scientific content as XRD and CECdata are not fully discussed and as these two technics do not give a proof of the formation of Zn-saponite.

line 106 : XRD diffractogram and not spectra.The description of the XRD patterns is poor.  For sample hot-plate -16, three reflections are observed and not two as the authors state. The authors assumed that a Zn-saponite is formed  but this can not be done without further characterizations. The determination of the phase purity lacks also of explaination and noting is said on the side phase(s) and especially on the presence of amorphous silica. The accuracy of the d001 and d060 value should be given.On lines 164, the authors claim that the (001) reflection of microwave 24-0.5E is stronger that that of Microwave 24 indicating favorable crystallinity. To me, this is not true as it seems that the two diffractohgrams are exactly the same.

as a conclusion, the proposed strategy to synthesize saponite is of interest but more characterization of the samples are needed to prove the formation of the desired phase and the paper has to be rewritten for clarity.

Reviewer 3 Report

This  paper describes the synthesis of Zn saponites using an new original microwave circulating reflux method. The paper needs major revisions for the Following reasons:

-1 The introduction is tendentious and the references not appropriate and up to date  . 

The sentence lines 45-48 is not correct: there is a lot of smectites syntheses below 100 °C (see a review in Klopproge et al., Clays Clay Min., 1999, 47, 529-544) even concerning Zn- saponite!(see Vogels et al., Amer. Min., 2005, 90, 931-944; Zhang et al., Appl. Clay sci., 2017, 135, 282-288) or stevensite (Petit et al.,2008 Clays and Clay Minerals 56, 645-654).

By an other way there is a lot of recent smectite syntheses using microwaves (see as exemple Baron et al., Current Miccowave Chem, 2016, 3, 85-89).

Lastly, Vincent et al. synthesised saponites (as said in the title of the paper) and not bentonites which are not smectites

 -2 Methods and results are not correctly presented:

Lines 77-83: what about  silica??

Figure 1: what was the temperature in the baker above the " temperature maintenance  device"?

Lines 97-98: What was the synhesis temperature using the hot plate?

Line 100: are they powder XRD patterns? What was the X-ray radiation?

Lines 102- 103: this method cannot be used to determine the phase purity. The CEC values of smectites dépends strongly to their hydration state. How were dehydrated the samples before CEC measurements?

Lines 106- 129:XRD data are not correctly given and discussed. Give the (hkl or hk) indices of diffraction peaks. To identify clearly a saponite a XRD trace after glycolation  is needed. Please give a figure with an experimental XRD pattern and its Rietveld simulation.

Line 127- 129 "critallinity" is inappropriate (amont of crystallised phase).

Lines 167-170: the "0.5- 1 hour synthesis time" is confusing. The true synthesis time was 24 hours during which the solid phase was exposed to microwave during only 0.57 hour!

Lines 171-192: The measured CEC values have two different origins: the first is due to the permanent layer charge, linked to the thedraedral substitutions for synthesised saponites and the second to surface charges. For many applications, the first one is only useful. Authors must calculate the amount of CEC due to the permanent layer charge of synthesised saponites according to their theoretical structural formulae, taking into account their hydratation state. The measered CEC values must be discussed then. The micowave synthesised saponites exhibiting a near 50% phase purity, their high CEC values must be dicussed.

Lastly, the precusor was maintained at a temperature above 20°C in the baker (Fig. 1). Consequently, the precursor may evolved also in the baker during the synthesis time. This needs to be discussed. 

Round 2

Reviewer 1 Report

The manuscript has been improved by the incorporation of the comments from the reviewers. It is almost ready to be accepted. However, small changes may before be done:

Line 133: The meaning of TACB abbreviation may be included. Please check other abbreviations, the meaning may be always included.

In lines 139 and 145, “spectrum” may be changed to “diffractogram” or “XRD pattern”.

English style may be still checked. I suppose that this may be done in the edition process. Before, the authors may carefully check the document. Here I include some corrections:

Lines 105-106: “In order to achieve significantly different of results” may be “In order to achieve significantly different results” or “In order to achieve significantly differences in the results”.

Line 158: “amont” may be “amount”.

Lines 215 and 219: “soponite” may be “saponite”.

Reviewer 2 Report

The manuscript has been significantly improved, there are now no more points to be adressed. I therefore recommend its publication after a last check of the english spelling.
